# Validation of a Human Challenge Model Using an LT-Expressing Enterotoxigenic *E. coli* Strain (LSN03-016011) and Characterization of Potential Amelioration of Disease by an Investigational Oral Vaccine Candidate (VLA1701)

**DOI:** 10.3390/microorganisms12040727

**Published:** 2024-04-03

**Authors:** Kawsar R. Talaat, Chad K. Porter, Subhra Chakraborty, Brittany L. Feijoo, Jessica Brubaker, Brittany M. Adjoodani, Barbara DeNearing, Michael G. Prouty, Steven T. Poole, A. Louis Bourgeois, Madison Billingsley, David A. Sack, Susanne Eder-Lingelbach, Christian Taucher

**Affiliations:** 1Department of International Health, Johns Hopkins Bloomberg School of Public Health, Baltimore, MD 21205, USA; schakr11@jhu.edu (S.C.); bjohns71@jhu.edu (B.L.F.); brittanymritko@gmail.com (B.M.A.); bdenear1@jhu.edu (B.D.); lbourge1@jhu.edu (A.L.B.); madison.i.m.billingsley@gmail.com (M.B.); dsack@jhu.edu (D.A.S.); 2Naval Medical Research Command, Silver Spring, MD 20910, USA; chad.k.porter2.civ@health.mil (C.K.P.); michael.g.prouty2.mil@health.mil (M.G.P.); spoole102@gmail.com (S.T.P.); 3Valneva, Austria GmbH, 1030 Vienna, Austria; susanne.eder-lingelbach@valneva.com (S.E.-L.); christian.taucher@valneva.com (C.T.)

**Keywords:** challenge study, enterotoxigenic *E. coli*, controlled human infection model, vaccine

## Abstract

Controlled human infection models are important tools for the evaluation of vaccines against diseases where an appropriate correlate of protection has not been identified. Enterotoxigenic *Escherichia coli* (ETEC) strain LSN03-016011/A (LSN03) is an LT enterotoxin and CS17-expressing ETEC strain useful for evaluating vaccine candidates targeting LT-expressing strains. We sought to confirm the ability of the LSN03 strain to induce moderate-to-severe diarrhea in a healthy American adult population, as well as the impact of immunization with an investigational cholera/ETEC vaccine (VLA-1701) on disease outcomes. A randomized, double-blinded pilot study was conducted in which participants received two doses of VLA1701 or placebo orally, one week apart; eight days after the second vaccination, 30 participants (15 vaccinees and 15 placebo recipients) were challenged with approximately 5 × 10^9^ colony-forming units of LSN03. The vaccine was well tolerated, with no significant adverse events. The vaccine also induced serum IgA and IgG responses to LT. After challenge, 11 of the placebo recipients (73.3%; 95%CI: 48.0–89.1) and 7 of the VLA1701 recipients (46.7%; 95%CI: 24.8–68.8) had moderate-to-severe diarrhea (*p* = 0.26), while 14 placebo recipients (93%) and 8 vaccine recipients (53.3%) experienced diarrhea of any severity, resulting in a protective efficacy of 42.9% (*p* = 0.035). In addition, the vaccine also appeared to provide protection against more severe diarrhea (*p* = 0.054). Vaccinees also tended to shed lower levels of the LSN03 challenge strain compared to placebo recipients (*p* = 0.056). In addition, the disease severity score was lower for the vaccinees than for the placebo recipients (*p* = 0.046). In summary, the LSN03 ETEC challenge strain induced moderate-to-severe diarrhea in 73.3% of placebo recipients. VLA1701 vaccination ameliorated disease severity, as observed by several parameters, including the percentage of participants experiencing diarrhea, as well as stool frequency and ETEC severity scores. These data highlight the potential value of LSN03 as a suitable ETEC challenge strain to evaluate LT-based vaccine targets (NCT03576183).

## 1. Introduction

Diarrhea is a leading global health problem, causing approximately 4% of all deaths worldwide and 5% of health loss due to disability. It is most frequently caused by gastrointestinal infections and is responsible for the deaths of around 1.6 million people each year, mostly affecting children in developing countries [1,2]. Diarrheal disease is caused by a host of bacterial, viral, and parasitic organisms, with many cases (80% in travelers) caused by bacteria that produce one or more enterotoxins [3,4]. Cholera, which results from infection with *Vibrio cholerae* bacteria, is the most severe of these diseases, whereas infection with enterotoxigenic *Escherichia coli* (ETEC) is a major cause of diarrheal disease in lower-income countries, especially among children [5,6]. In addition, ETEC is the leading cause of diarrhea among international travelers and deploying military personnel [7,8].

ETEC mortality estimates range between 18,700 deaths [2,9] (Institute for Health Metrics and Evaluation (IHME) estimates) and 42,000 deaths (maternal child epidemiology (MCEE) estimates) [10] in children younger than 5 years. Among those older than five years, the IHME estimated 32,084 annual deaths, with most occurring in South Asia and Africa [11]. While diarrhea mortality rates for ETEC and other enteric pathogens are declining, there are not the same declines in morbidity due to diarrhea. ETEC, along with *Shigella* and *Salmonella*, is associated with higher case fatality rates in infants under 1 year of age and among hospitalized infants and children [12], as well as an increased mortality rate in children under 24 months. Among older age groups (>5 years of age), both ETEC and cholera remain important causes of potentially life-threatening watery diarrhea, and they are frequent contributors to recent outbreaks of cholera-like illness in South Asia and Africa [13,14,15].

Despite the decrease in diarrheal morbidity, vaccine development for ETEC continues to be a priority [16,17]. This is partially fueled by the overuse of antibiotics to treat diarrheal infections and the rise of antimicrobial resistance (AMR) in Enterobacteriaceae, with the spread of resistant ETEC and *V. cholerae* making treatment harder when indicated [18]. The need for a vaccine against ETEC is further strengthened by the accumulation of increasing evidence that shows that both symptomatic and asymptomatic ETEC infections, along with other diarrheal infections, contribute to childhood stunting and gut enteropathy [19,20,21,22]. Significant elevations in intestinal biomarkers of inflammation are seen in ETEC controlled human infection model (CHIM) studies in both symptomatic and asymptomatic participants. These data are also supported by recent in vitro studies demonstrating that the heat-labile toxin (LT) of ETEC can drive several modulators of enteropathic changes in the small-intestinal epithelia [23].

Specific virulence factors such as LT and heat-stable toxin (ST) enterotoxins and colonization factors (CFs) differentiate ETEC from other diarrheagenic *E. coli* [5]. ETEC colonizes the small intestine by virtue of CFs, followed by the secretion of heat-stable toxin (ST) and/or heat-labile toxin (LT), leading to secretory watery diarrhea. The LT produced by ETEC is structurally, pathophysiologically, and immunologically similar to cholera toxin (CT). The quaternary structures of LT and CT are also almost identical, each with one A subunit surrounded by a pentamer of B subunits, which bind to the GM1 ganglioside receptor [24,25]. Most antibodies induced by CT are directed against the B subunit [25], and cholera toxin subunit (CTB)-specific antibodies cross-react with LT [3]. Sera from individuals vaccinated with CT neutralize LT [26], and individuals infected with V. cholerae through challenge or natural infection show an increase in cross-reacting antitoxin antibodies, suggesting similar antigenic characteristics of both toxins [27]. The immune response to CTB also neutralizes the toxic effect of LT and is associated with protection against some forms of ETEC-associated diarrhea in the field [27,28,29,30,31]. Anti-LT reactive antibodies modulate the severity of ETEC-associated illness in a CHIM [31], protect against ETEC strains producing only LT toxin in field studies, and reduce the severity of ETEC-associated disease in general in the field [31,32].

Efforts to accelerate ETEC vaccine development are increasingly turning to CHIMs as an early assessment of protective efficacy to advance the most promising candidates into more expanded development and field testing. Well-characterized ETEC CHIM strains exist for many common phenotypes, while challenge strains to assess the effectiveness of a vaccine against ETEC strains making only LT or ST are only now being developed [33,34]. The most advanced LT-only strain suitable for CHIMs is LSN03-016011/A (LT+ and CS17+); however, this strain has only been given to 48 participants (25 naïve), at doses ranging from 5 × 10^8^ to 5 × 10^9^ cfu. Additional evaluation of this strain in CHIMs is needed to better define the incidence of moderate-to-severe diarrhea and further confirm its suitability for testing vaccine candidates. Therefore, we designed a CHIM with the primary objective of further defining the spectrum of clinical illness resulting from infection with the LSN03-016011/A strain. A secondary objective (for which this study was not powered) was to assess the efficacy of a candidate cholera/ETEC vaccine (VLA1701) in reducing the incidence and severity of enteric illness. The results of these studies will help determine whether the LSN03-016011/A strain is suitable for playing an effective role in future cholera and ETEC vaccine-driven disease control efforts, particularly for selection of the best vaccine candidates for more advanced clinical development. The need for simpler, more effective cholera vaccines is attracting increasing international attention, since reducing the cost of goods might help expand their availability and use. The dual market potential of OCVs like VLA1701, with possible indications for use in both LMICs and international travelers as a preventive intervention against both cholera and ETEC, also serves to potentially increase the vaccine’s value proposition and could further incentivize production.

## 2. Materials and Methods

### 2.1. Clinical Trial Design

This was a single-center, randomized, double-blind, placebo-controlled pilot study designed to further define the spectrum of clinical illness induced by the LSN03-016011/A ETEC strain in a CHIM study. Initially, 34 participants were randomized 1:1 to receive two doses of either VLA1701 or placebo orally, 7 days apart, and followed as outpatients for safety and to assess vaccine immunogenicity. Approximately 7 days after the second dose, 30 participants were selected for ETEC challenge based on eligibility (see Figure 1 and Appendix A).

### 2.2. Study Oversight

This study was conducted at the Johns Hopkins Bloomberg School of Public Health (JHBSPH) Center for Immunization Research (CIR), and subjects were challenged at the CIR inpatient unit at the Johns Hopkins Bayview Medical Campus. All participants provided written informed consent. The trial was approved by the JHBSPH Institutional Review Board, in compliance with all federal regulations governing the protection of human participants. Valneva served as the sponsor of the study and developed the study design with the investigators. The investigators were responsible for the study’s conduct, management, and data analysis (ClinicalTrials.gov Identifier: NCT03576183).

### 2.3. Investigational Vaccine

The investigational vaccine VLA1701 used for this pilot CHIM is an oral inactivated vaccine containing inactivated *V. cholerae* and 1 mg of recombinant cholera toxin B subunit (rCTB). VLA1701 shares some features with an oral killed whole-cell vaccine against cholera caused by *V. cholerae* O1 (classical and El Tor biotypes) [35]. However, VLA1701 contains fewer killed *V. cholerae* strains (2 instead of 4). VLA1701 does contain an increased number of *V. cholerae* O1 bacteria of the El Tor biotype, since this is the causative organism of the 7th cholera pandemic. The formulation of VLA1701 is shown below.

A total of 1.25 × 10^11^ bacteria of the following strains:*Vibrio cholerae* O1 Inaba, El Tor biotype (formalin-inactivated), 6.25 × 10^10^ vibrios;*Vibrio cholerae* O1 Ogawa, classical biotype (heat-inactivated), 6.25 × 10^10^ vibrios;Recombinant cholera toxin B subunit (CTB), 1 mg.

The VLA1701 vaccine is administered orally in 2 doses at least 1 week apart. Subjects must fast for 1 h before and after each dose. The vaccine is provided as 2 components: a liquid suspension of the inactivated vibrio whole cell in a glass vial, along with a sachet of effervescent granules containing sodium hydrogen carbonate buffer. The effervescent granules are dissolved in 150 mL of bottled water before adding the contents of the vaccine vial. The placebo was just the effervescent granules dissolved in 150 mL of bottled water. The study participants drank the vaccine or placebo from opaque cups.

### 2.4. ETEC Challenge Strain Characteristics and Preparation of Challenge Dose for Administration to Participants in the Trial

LSN03-016011/A (LT+, ST−, CS17+) is a well characterized ETEC strain manufactured as a frozen production cell bank under cGMP by the Walter Reed Army Institute of Research (Silver Spring, MD, USA). This strain has been used in two prior human clinical trials [34,36]. The target challenge dose for this trial was 5 × 10^9^ cfu. The challenge inoculum was prepared from fresh plate-grown organism as described previously. Expression of CS17 was confirmed by agglutination in anti-CS17 rabbit antiserum prior to administration to the study participants. The challenge inoculum was diluted in sodium bicarbonate buffer (13.35 g of NaHCO_3_/L) (Humco, Austin, TX, USA). Each participant drank 120 mL of plain sodium bicarbonate buffer 1 min prior to ingesting the LSN03-016011/A challenge inoculum in 30 mL of the same buffer.

### 2.5. Study Population and Enrollment Criteria

Eligible participants included healthy non-pregnant adults recruited from the Mid-Atlantic region, between 18 and 50 years of age, with no significant medical conditions. Participants were ineligible if they had previously been exposed to cholera or ETEC, including the receipt of LT, in the last 3 years, as assessed by medical history, travel history, and potential employment-based exposure. Informed consent was a rigorous and iterative process to ensure comprehension of the trial and their participation. To ensure that the eligibility criteria were met, medical history, laboratory tests, and a complete physical exam were performed. Participants with childbearing potential completed a pregnancy test before each vaccination and prior to receiving the challenge. A full list of inclusion and exclusion criteria can be found in the Appendix A.

### 2.6. Study Procedures

Participants were vaccinated (2 doses, 1 week apart) with VLA1701 or placebo as a single group. Approximately 7 days later, 30 participants were admitted to the inpatient unit. The ETEC CHIM procedures have previously been published [34,36,37]. Briefly, the morning following admission, the participants were challenged with the LSN strain after a 90-min fast. The participants fasted for 90 additional minutes following the challenge. After the challenge, the participants were monitored for diarrhea and other signs/symptoms of enteric illness by daily medical checks, vital signs at least thrice daily, and grading and weighing of all stools. Symptoms of ETEC were expected to range from mild to severe watery diarrhea and possibly include nausea, vomiting, abdominal cramping, headache, abdominal gurgling or gas, anorexia, fever, muscle and/or joint aches, and malaise. Participants experiencing loose stools were provided oral rehydration and closely monitored for hypovolemia. They were treated with intravenous fluids (IVFs) as necessary. Five days after the challenge, or sooner if the participants met early treatment criteria, the participants were treated with antibiotics (ciprofloxacin 500 mg twice daily for 3 days). Participants were discharged after at least two doses of antibiotics, clinical symptoms had resolved or were resolving, and the participant had produced two stool samples that were negative for LSN03-016011/A by culture. All participants had an in-person visit 28 days after the challenge and a telephone call to check for any serious medical conditions, new onset of chronic illnesses, and functional bowel disorders approximately 6 months after their first vaccination. The study design details and study participants’ allocation consort diagram are provided in Figure 1 and Appendix A.

### 2.7. Study Diarrhea Definitions and Criteria for Antibiotic Treatment

All stools were graded 1–5 as previously described in [38], and those graded as grade 3, 4, or 5 were considered ‘loose stools’ and could contribute to an episode of diarrhea:Severe diarrhea (six loose stools or >800 g in 24 h;Moderate diarrhea: four to five loose stools or 401 to 800 g in 24 h;Mild diarrhea: one to three loose stools or ≤400 g in any 24 h period.

An episode of diarrhea was considered complete after 24 h without a loose stool. The criteria for early antibiotic treatment prior to 120 h post-challenge were as follows:Severe diarrhea (6 or more loose stools or >800 g in 24 h);Stool output consistent with moderate diarrhea for 48 h;Mild or moderate diarrhea and two or more of the following symptoms: severe abdominal pain, severe abdominal cramps, severe nausea, severe headache, severe myalgia, any fever (≥38.0 °C), or any vomiting.

A study physician determined that early treatment was warranted for any reason.

### 2.8. Safety

The primary objective was to estimate the percentage of participants with moderate-to-severe diarrhea within 120 h of challenge with ETEC strain LSN03-016011/A. Secondary endpoints included calculation of the severity of disease induced after challenge using the ETEC disease score [39], as well as the safety of the VLA1701 vaccine as measured by the percentage of participants with solicited adverse events (AEs) within 7 days after each vaccination. The ETEC disease severity score was developed from prior controlled human ETEC infection studies. Data from these studies were used to derive the three-parameter disease severity (including stool output and clinical signs and symptoms). Correlation of univariate regression across sign and symptom severity from these prior trials was performed. A multiple correspondence analysis was conducted, and the 3-parameter disease score was developed and validated by comparison to standard outcome definitions and applied to prior ETEC challenge trials [39]. Additional analyses were performed to assess the incidence of all AEs (including SAEs) prior to challenge, and to estimate the percentage of participants with any AEs or SAEs or any investigational medicinal product (IMP)-related AEs or SAEs during the entire study period.

Additional exploratory endpoints for the trial included further evaluation of the spectrum of disease induced by infection with the LSN03-016001/A strain, either the total number or volume of loose stools, the time to onset and duration of diarrhea, or the number of participants receiving IVFs or requiring early antibiotics. Also, the number of colony-forming units (cfu) of ETEC shed in the stool after challenge was calculated on days 2 and 4 post-challenge.

### 2.9. Immunogenicity Assessment

The immunogenicity of VLA1701 was determined by analyzing systemic and mucosal immune responses at baseline and one week after receipt of the second vaccine dose. Systemic and mucosal immunity was also assessed before, one week after, and one month after challenge with ETEC strain LSN03-016011/A. Immunogenicity was determined by assessing serum IgG and IgA geometric mean titers (GMT) for CT (cholera toxin B subunit). Among those in the challenge population, serum IgG and IgA were analyzed by timepoint, and the absolute change was compared to the samples gathered at the first vaccination visit. A 2-fold change from baseline was considered to indicate a response to the vaccine. Additional analyses were performed to test for seroconversion prior to challenge and IgA responses against LT and CS17 antibodies in lymphocyte supernatants (ALS), as well as mean vibriocidal antibodies. Immunogenicity analyses also evaluated fecal IgA against LT and CS17 using previously published methods [34,36,40,41,42]. These analyses are not included in this manuscript and will be included in a subsequent immunology paper.

### 2.10. Microbiological Assessment

At least one stool sample per participant per day (or a rectal swab if a sample was not produced) was cultured. As previously described, qualitative stool cultures were performed daily, and quantitative cultures performed 2 and 4 days after challenge to assess differences in transit time and colonization between the study groups [34,36]. Non-lactose-fermenting colonies on MacConkey agar were screened for CS17 production on CFA agar with bile salts by the colony immunoblot method, using anti-CS17 antisera to quantitate shedding of the ETEC LSN03-016011/A strain, as previously described [34,36].

### 2.11. Statistical Analysis

The proportion of subjects meeting the primary endpoint (moderate-to-severe diarrhea as assessed by the Independent Outcome Adjudication Committee) was compared between the two treatment groups using Fisher’s exact test, and two-sided 95% confidence intervals (CIs) were estimated [43]. This was performed for any diarrhea and other categorical clinical endpoints. All continuous efficacy parameters (excluding the disease severity score) were compared between treatment groups with the Wilcoxon-Mann-Whitney test. The disease score was compared between groups by using the Cochran-Armitage test of trends (the trend being the natural order of the ordinal score). For secondary comparisons, Student’s *t*-test was applied to the scores. Geometric means of immunogenicity endpoints were accompanied by a 95% CI. All AE rates (excluding rates on the system organ class [SOC] and preferred term [PT] level) were compared between the two groups using a two-sided Fisher’s exact test. Adverse event rates were accompanied by two-sided 95% CIs [43]. Differences between IgG and IgA levels were compared using Student’s *t*-test. All tests were performed under a two-sided alpha = 0.05. Prism version 10.2.2 was used for statistical analyses (GraphPad Software, Boston, MA, USA).

### 2.12. Role of Funding Source

The funder of this study (Valneva) helped to design the study but had no role in data collection, data analysis, data interpretation, or the writing of the report. The corresponding author had full access to all of the data in the study and had final responsibility for the manuscript and the decision to submit for publication.

## 3. Results

### 3.1. Study Population

As shown in the diagram below (Figure 1), a total of 42 people were screened for the study, with 34 individuals enrolled, randomized to the vaccine and placebo groups, and included in the safety/intention-to-treat population. Among the placebo group, one person withdrew prior to receipt of the second dose. Of the 33 who received both vaccine and placebo doses, 30 were enrolled in the challenge phase. The study timeline detailing the major study-related procedures, from study initiation through study close-out, is shown in Appendix A. The study population was predominantly male (54.8%) and African-American (69%), with a mean age of 34.5 years. Additional demographic characteristics of the study population are outlined in Appendix A. The participants randomized to the vaccine and placebo groups had similar demographic characteristics (Appendix A).

### 3.2. Challenge with LSN03-016011: Attack Rate and Symptoms, and Impact of VLA1701 on the Challenge

Following challenge with 1 × 10^10^ cfu of the LSN03-016011 ETEC strain, the placebo group had an attack rate of 73.3% for moderate-to-severe diarrhea (see Table 1 below), consistent with prior studies using this strain [34,36]. In contrast, only 46.7% of the vaccine recipients developed moderate-to-severe diarrhea (*p* = 0.26). Rates of moderate-to-severe diarrhea did not vary significantly by ABO blood type. The proportion of placebo recipients with diarrhea of any severity was significantly higher (93.3%) than in the vaccine recipients (53.3%) (*p* = 0.04). Significantly fewer vaccinees (13.3%) also experienced severe diarrhea compared to the placebo recipients (46.7%) (*p* = 0.054). None of the VLA1701 vaccine recipients required IVFs, whereas IVFs were required for five (33.3%) of the placebo recipients (*p* = 0.04). Approximately twice as many placebo recipients required early antibiotic treatment compared to the vaccinees (53.3% and 26.7%, respectively); however, this difference was not significantly significant, as the study was not powered for this (*p* = 0.26) (Table 1).

The median stool output among vaccinees and placebo recipients developing diarrhea (both stool number and volume) is shown in Table 2 below. While the stool output tended to be higher in placebo recipients, there were no statistically significant differences. In participants with moderate/severe diarrhea, a lower number of loose stools was observed in the VLA1701 group (mean 9.4) than in the placebo group (mean 12.2) (*p* = 0.0492). Gastrointestinal and systemic signs and symptoms also did not differ significantly between the vaccine and placebo recipients (see Figure 2a).

A lower median ETEC disease severity score was observed for the VLA1701 recipients (2.0) than for the placebo recipients (4.0) (*p* = 0.0774) (Figure 2b). Additionally, estimates of vaccine efficacy increased with increasing disease severity cutoff points (see Figure 3a below). When the disease severity score was evaluated as a dichotomous outcome, 9 of 15 (60%) of the placebo participants had a score of ≥4 (the median disease severity score in the placebo group), while among the vaccinees only 3 of 15 (20%) had a score of ≥4, which is a protective efficacy against a severity score of ≥4 of 66.6% (95% CI 0.5–88.8%; *p* = 0.03). A similar trend was noted for the vaccine’s impact on higher scores (protective efficacy (PE) > 50%) (see Figure 3), but these values were not significant due to the sample size limitations of this study.

Vaccinees tended to shed lower levels of the LSN03-016011 strain at days 2 and 4 post-challenge, as compared with placebo recipients. On day 2, the mean cfu/g of stool was 1.28 × 10^7^ for the VLA1701 group and 2.11 × 10^7^ for the placebo group (*p* = 0.06). On day 4 post challenge, the mean cfu/g of stool value was 9.17 × 10^6^ for the VLA1701 group and 1.15 × 10^7^ for the placebo group (*p* = 0.9), although several participants had already been treated (Figure 3b).

### 3.3. Safety Assessment of Oral Immunization with the VLA1701 Vaccine

Overall, the VLA1701 vaccine was well tolerated. The most frequently reported solicited AEs by symptom were diarrhea and headache, which were both experienced by three (17.6%) participants in the VLA1701 treatment group and two (11.8%) participants in the placebo group. Chills, abdominal cramping, fatigue, and rash were reported in the placebo group, but not in the VLA1701 treatment group (Appendix A). All solicited Aes were mild or moderate; one participant in the vaccine group developed moderate diarrhea after vaccination.

Two unsolicited Aes in the VLA1701 treatment group (abnormal gastrointestinal sounds and thirst) and one in the placebo group (hot flashes) were felt to be at least possibly related to the treatment. There were no differences in the solicited or unsolicited adverse events between the vaccine and placebo groups. No SAEs, Aes of special interest, or Aes leading to withdrawal from the study or from further vaccination were reported during the study period. Overall, there were no statistically significant differences between the treatment groups in observations of any of these categories of Aes (overall, related, severe, related-severe, leading to withdrawal from vaccination, or Aes of special interest).

### 3.4. Immunogenicity of the VLA1701

Vaccination with VLA1701 induced IgG responses to CT in 81.3% of vaccinees and IgA responses in 75% of vaccinees. No placebo recipients mounted serum IgA or IgG responses to CT. Among the 17 participants given two doses of VLA1701, 14 (82.4%) vaccinees mounted ≥4-fold rises in their SBA titers following immunization. The kinetics of the serum IgA and IgG antibodies to CT is shown in Figure 4.

Following challenge, vaccinated participants showed greater increases in serum IgA (6.9-fold rise) and IgG (11.4-fold rise) to CT over baseline than the placebo recipients, indicating that ETEC challenge triggered a booster response in these individuals. Possible relationships between immune response frequencies and levels post-immunization and clinical outcomes following challenge with the LSN03-016011 ETEC strain will be addressed in a subsequent manuscript.

## 4. Discussion

The primary objective of this study was to confirm the clinical aspects of disease following inoculation with the LSN03-016011 ETEC challenge strain. Prior to this study, the LSN03-016011/A strain challenge had been administered to a total of 48 human participants, only 25 of whom were naïve, with doses ranging from 5 × 10^8^ to 5 × 10^9^ cfu [34,36], so data with this strain are limited. In the 20 naïve participants who had been challenged with 5 × 10^9^ cfu previously, 65% developed diarrhea of any severity and 50% developed moderate-to-severe diarrhea. In our study, 93.3% of placebo recipients had diarrhea of any severity and 73.3% had moderate-to-severe diarrhea after challenge with 1 × 10^10^ cfu. Despite the differences in attack rate with historical studies, the median time to diarrhea onset in our placebo recipients was 20 h, similar to what was previously reported (21.7 h) [36]. We also observed similar number (9.0) and total volume of loose stools (953.5 g) compared to prior studies (8.0 and 910 g, respectively) [37]. The ETEC severity score among placebo recipients in our study was also higher (median 4; range 0.0–6.0) than has been historically observed for the LSN03-016011/A strain. Compared to other ETEC CHIM strains such as H10407, B7A, and E24377A, the LSN03-016011/A strain presented with a lower ETEC severity score [37].

We observed a significant reduction in diarrhea of any severity among VLA1701 recipients (*p* = 0.04), highlighting the potential value of this model as a tool in future ETEC vaccine development efforts, as well as the potential efficacy of VLA1701 against LT-expressing ETEC. A limitation of this study is its small sample size, as we were limited by the capacity of the inpatient facility. While VLA1701 decreased all diarrhea, it did not significantly decrease the incidence of moderate-to-severe diarrhea, as the sample size for the study was not based on this comparison and was underpowered to demonstrate a significant difference in rates. However, we did observe a significantly lower ETEC disease severity score among the VLA1701 vaccinees than in the placebo recipients. Second, only 13.3% of participants in the VLA1701 experienced severe diarrhea, compared with 46.7% in the placebo group. Third, no participants in the VLA1701 group required IV fluids, compared with 33.3% in the placebo group. Fourth, half as many vaccinees met the criteria for early treatment (26.7%) as did placebo recipients (53.3%). Lastly, even those with moderate/severe diarrhea in the VLA1701 group had fewer loose stools compared with those in the placebo group. Consistent with our observations that immunization with VLA1701 reduced the incidence and severity of the enteric illness caused by the LSN03-01611 ETEC strain, when the disease severity score ≥ 4 was evaluated as a dichotomous efficacy endpoint, vaccination withVLA1701 was 66.7% efficacious.

The encouraging observations with a vaccine candidate containing rCTB, despite the small sample size, were not unexpected, since the immune response to CTB has also been shown to neutralize the toxic effect of the ETEC LT and to be associated with protection against some forms of LT-ETEC-associated diarrhea in the field [3,28,29]. Anti-LT reactive antibodies have also been shown to modulate the severity of ETEC-associated illness in a CHIM [31], to protect against ETEC strains producing only LT in field studies, and to reduce the severity of ETEC-associated disease in general in the field [31,32,44]. Similarly, higher anti-CTB serum IgA titers are associated with a reduced risk of ETEC-attributable travelers’ diarrhea and lower disease severity scores in U.S. travelers to Guatemala and Mexico [30,31,45]. This study also provides the first information on the systemic and mucosal immunogenicity of VLA1701. In the absence of a surrogate of protection, this study aimed to identify whether serum IgA and IgG antibody responses and two measures of mucosal immunogenicity (ALS and/or fecal IgA antibody responses to VLA1701 antigens) were associated with reduced disease following challenge.

VLA1701 immunization induced a robust immune response against the CT toxin. The strong serum IgA and IgG anti-CT responses induced in this trial by the vaccine are supportive of its positive impact on ETEC-associated illness in this challenge study. Further analysis of the secondary and exploratory endpoints investigating the immunogenicity and safety of VLA1701 also supported the further development of an rCTB-containing vaccine candidate and highlighted the important additional benefit of this vaccine approach given its impact on ETEC disease incidence and severity in this rigorous CHIM. VLA1701 was immunogenic as measured by serum IgG and IgA levels against CT toxin. The vaccine-induced immune titers were greater after vaccination than those induced in the placebo group after challenge. There were no significant differences between the groups with regards to solicited adverse events, unsolicited adverse events, or unsolicited adverse events related to the vaccination. In addition, there were no serious adverse events observed during the entire study period.

## 5. Overall Conclusions

The observed attack rate of 73.3% for moderate-to-severe diarrhea, along with the median disease severity score of 4 in the placebo group, confirmed the previously observed attack rate for the challenge strain LSN03-016011/A, further highlighting the suitability of this ETEC strain as a tool in the development of active and passive preventive and treatment interventions for ETEC-induced diarrhea. Despite the small sample size, VLA1701-vaccinated subjects were less likely to experience any ETEC-induced diarrhea and had less severe disease when illness occurred compared to the placebo recipients. VLA1701 was safe and induced both mucosal and systemic immune responses against CT.

## Figures and Tables

**Figure 1 microorganisms-12-00727-f001:**
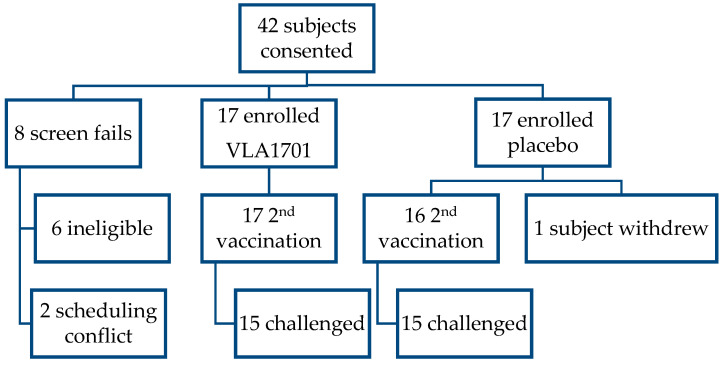
Overview of the enrollment, vaccination, and challenge phases of the study.

**Figure 2 microorganisms-12-00727-f002:**
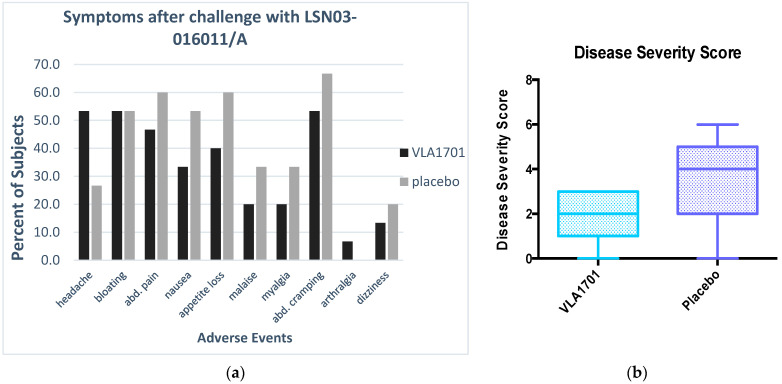
Comparison of associated symptoms (**a**) and median disease severity scores (**b**) in the vaccine and placebo groups following ETEC challenge. Note: Individual signs and symptoms and disease severity scores did not differ significantly between vaccine and placebo recipients.

**Figure 3 microorganisms-12-00727-f003:**
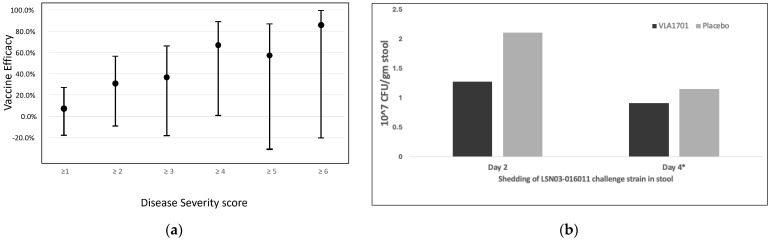
VLA1701 demonstrated increased efficacy (nonsignificant) against more severe disease after challenge with LSN03-016011 (**a**). Vaccinees tended to shed lower levels of the challenge strain compared with placebo recipients (**b**). * Only participants who developed diarrhea are included.

**Figure 4 microorganisms-12-00727-f004:**
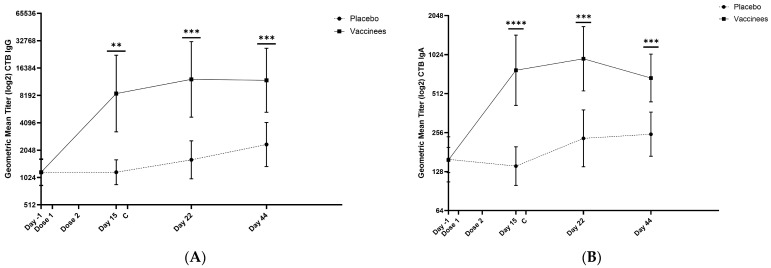
Kinetics of serum IgG (**A**) and IgA (**B**) antibody responses to CT after receipt of VLA1701. Note: The vaccine and placebo recipients contributing anti-CT titers to these figures were challenged with 5 × 10^9^ CFU of the LSN03-016011/A ETEC strain on day 16 (~one week after the second dose of the VLA1701 vaccine or placebo). C = challenge. CTB = cholera toxin B. ** *p*< 0.01; *** *p* < 0.001; **** *p* < 0.0001.

**Table 1 microorganisms-12-00727-t001:** Summary of diarrhea outcomes and intervention with IV fluids and early antibiotic treatment among vaccine and placebo recipients challenged with 5 × 10^9^ cfu of the LSN03-016011 ETEC strain.

Participants Experiencing Diarrhea *n* (%) [95% CI]	VLA1701 (*n* = 15)	Placebo (*n* = 15)	*p*-Value
Moderate-to-severe diarrhea	7 (46.7) [24.8, 69.9]	11 (73.3) [48.0, 89.1]	0.264
Any diarrhea	8 (53.3) [30.1, 75.2]	14 (93.3) [70.2, 98.8]	0.035
Mild diarrhea	1 (6.7) [1.2, 29.8]	3 (20.0) [7.0, 45.2]	0.598
Moderate diarrhea	5 (33.3) [15.2, 58.3]	4 (26.7) [10.9, 52.0]	1.000
Severe diarrhea	2 (13.3) [3.7. 37.9]	7 (46.7) [24.8, 69.9]	0.109
Received intravenous fluids	0 (0.0) [0.0, 20.4]	5 (33.3) [15.2, 58.3]	0.042
Early antibiotic treatment	4 (26.7)	8 (53.3)	0.264

**Table 2 microorganisms-12-00727-t002:** Diarrheal stool volume and quantity after challenge.

Median Diarrheal Stool Output (Interquartile Range)	VLA1701 (*n* = 8)	Placebo (*n* = 14)	*p*-Value
Total weight (g)	746 (579.0–1011)	953.5 (643–1349)	0.609
Total number	6.5 (6.0–8.5)	9 (7.0–11)	0.149
Maximum 24-h volume (g)	499.0 (323.0–794.0)	553.5 (358.0–907.0)	0.919
Maximum 24-h number	4.0 (3.5–7.0)	5.0 (4.0–7.0)	0.512

## Data Availability

Data is available upon request.

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
