# Peer review of "Validation of a Human Challenge Model Using an LT-Expressing Enterotoxigenic E. coli Strain (LSN03-016011) and Characterization of Potential Amelioration of Disease by an Investigational Oral Vaccine Candidate (VLA1701)"

_microorganisms, 2024, doi:10.3390/microorganisms12040727_

Round 1

Reviewer 1 Report

Comments and Suggestions for Authors

This review clearly described the immunogenicity of VLA 1701 and detailed challenge results with Enterotoxigenic Escherichia coli (ETEC) strain and confirmed that controlled human infection models are useful tools for the evaluation of vaccines against diseases when the suitable biomarker of protection is not available. Their reliable data proved that, despite the small sample size, VLA1701-vaccination could effectively prevent severe ETEC diarrhea and reduce disease severity compared to placebo recipients, indicating that VLA1701 is a promising safe vaccine to elicit both mucosal and systemic immunity against CT. This paper is helpful for further successful development of vaccine to control ETEC disease. Therefore, this manuscript is acceptable for publication after minor revisions.

1.     There are some careless grammatical mistakes and wrong use of word in line 229 and 335, which should be corrected.

2.     The result of fecal IgA against LT and CS17 should be added besides the IgA in the sera to show the better interesting comparison between the systemic and mucosal humoral immune responses induced by the VLA1701.

Author Response

Comments and Suggestions for Authors

This review clearly described the immunogenicity of VLA 1701 and detailed challenge results with Enterotoxigenic Escherichia coli (ETEC) strain and confirmed that controlled human infection models are useful tools for the evaluation of vaccines against diseases when the suitable biomarker of protection is not available. Their reliable data proved that, despite the small sample size, VLA1701-vaccination could effectively prevent severe ETEC diarrhea and reduce disease severity compared to placebo recipients, indicating that VLA1701 is a promising safe vaccine to elicit both mucosal and systemic immunity against CT. This paper is helpful for further successful development of vaccine to control ETEC disease. Therefore, this manuscript is acceptable for publication after minor revisions.

  1. There are some careless grammatical mistakes and wrong use of word in line 229 and 335, which should be corrected.

We thank the reviewer for their careful review, we have fixed these errors.

  1. The result of fecal IgA against LT and CS17 should be added besides the IgA in the sera to show the better interesting comparison between the systemic and mucosal humoral immune responses induced by the VLA1701.

The additional immunological assessments will be included in a subsequent paper.

Reviewer 2 Report

Comments and Suggestions for Authors

This manuscript presents a pilot study investigating LT-expressing ETEC strain LSN03-016011 as a challenge model in volunteers vaccinated (or unvaccinated) with oral ETEC vaccine candidate (VLA1701). Controlled human infection models (CHIMs), such as described here, are important is assessing vaccine efficacy in diseases without clear correlates of protection. This study adds to previous small-scale studies assessing LSN03.

The paper is well written and data are generally clearly presented. The research objectives are clear and the design of the study is appropriate to address these. 

Specific comments / suggestions.

1. Should the title not be altered to indicate that VLA1701 is a cholera/ETEC TL vaccine? As it stands it appears that the vaccine was designed as an ETEC vaccine, however it is primarily a cholera vaccine which has cross-protection with ETEC.

2. Around line 108/110 there are some formatting/editing issues: vac-cine instead of vaccine and The need....are getting attention instead of is getting attention.

2. In the methodology, the vaccine formulation is presented with open and closed bullets? Not sure if this is a formatting error and if not it needs clarification.

3. The VLA1701 vaccine is an oral vaccine, yet no detail is provided on whether it was taken as a syrup, pill, powder or other formulation? Similarly, the challenge formulation administered is not clear.

4. The methodology should include a section on disease severity scoring. Data are presented using this and the only access to any explanation is provided in supplementary material.

5. Figures 2 and 3 should include statistics, and if nothing is significant a mention of this in the legend. 

6. Figure 4 should also include statistics and an indication of sample sizes used for determination of median serum IgG and A levels.

The overall conclusion that the attack rate of the challenge strain was >73%, and agrees with previous studies that this is a valid model is sound. Inclusion of the immune data post-challenge would have been interesting and added to the study (however this to be included in a separate article).

Author Response

Comments and Suggestions for Authors

This manuscript presents a pilot study investigating LT-expressing ETEC strain LSN03-016011 as a challenge model in volunteers vaccinated (or unvaccinated) with oral ETEC vaccine candidate (VLA1701). Controlled human infection models (CHIMs), such as described here, are important is assessing vaccine efficacy in diseases without clear correlates of protection. This study adds to previous small-scale studies assessing LSN03.

The paper is well written and data are generally clearly presented. The research objectives are clear and the design of the study is appropriate to address these. 

We thank the reviewers for their kind words.

Specific comments / suggestions.

  1. Should the title not be altered to indicate that VLA1701 is a cholera/ETEC TL vaccine? As it stands it appears that the vaccine was designed as an ETEC vaccine, however it is primarily a cholera vaccine which has cross-protection with ETEC.

We have retitled the paper to remove the ETEC designation for the vaccine

  1. Around line 108/110 there are some formatting/editing issues: vac-cine instead of vaccine and The need....are getting attention instead of is getting attention.

Thank you, these have been fixed.

  1. In the methodology, the vaccine formulation is presented with open and closed bullets? Not sure if this is a formatting error and if not it needs clarification.

This was an error, it has been fixed.

  1. The VLA1701 vaccine is an oral vaccine, yet no detail is provided on whether it was taken as a syrup, pill, powder or other formulation? Similarly, the challenge formulation administered is not clear.

Additional information about both the vaccine (section 2.3) and the challenge (section 2.4) was added to the manuscript.

  1. The methodology should include a section on disease severity scoring. Data are presented using this and the only access to any explanation is provided in supplementary material.

The disease severity score details have been added to section 2.9.

  1. Figures 2 and 3 should include statistics, and if nothing is significant a mention of this in the legend. 

This has been added to the Figures.

  1. Figure 4 should also include statistics and an indication of sample sizes used for determination of median serum IgG and A levels.

This has been added to the figures. All of the volunteers who were challenged were included in the determination of antibodies.

The overall conclusion that the attack rate of the challenge strain was >73%, and agrees with previous studies that this is a valid model is sound. Inclusion of the immune data post-challenge would have been interesting and added to the study (however this to be included in a separate article).

Reviewer 3 Report

Comments and Suggestions for Authors

The manuscript by Talaat et al., reported the results of the randomized, double-blinded pilot study in which participants received 2 doses of VLA1701 or placebo orally one week apart and then were challenged with LSN03. The primary objective of the study was to estimate the percentage of participants with moderate to-severe diarrhea.

The protocol of the study and the results are clearly described and support the potential efficacy of VLA1701 in the prevention of diarrhea diseases. It would be interesting to know how long elicited antibodies persist in the serum.

Author Response

Comments and Suggestions for Authors

The manuscript by Talaat et al., reported the results of the randomized, double-blinded pilot study in which participants received 2 doses of VLA1701 or placebo orally one week apart and then were challenged with LSN03. The primary objective of the study was to estimate the percentage of participants with moderate to-severe diarrhea.

The protocol of the study and the results are clearly described and support the potential efficacy of VLA1701 in the prevention of diarrhea diseases. It would be interesting to know how long elicited antibodies persist in the serum.

Unfortunately, as we challenged nearly everyone who vaccinated, we were unable to assess duration of antibodies.